# High-Throughput Sequencing of Complementarity Determining Region 3 in the Heavy Chain of B-Cell Receptor in Renal Transplant Recipients: A Preliminary Report

**DOI:** 10.3390/jcm11112980

**Published:** 2022-05-25

**Authors:** Tsai-Hung Wu, Hsien-Tzung Liao, Tzu-Hao Li, Hung-Cheng Tsai, Niang-Cheng Lin, Cheng-Yen Chen, Shih-Feng Tsai, Tzu-Hao Huang, Chang-Youh Tsai, Chia-Li Yu

**Affiliations:** 1Division of Nephrology, Taipei Veterans General Hospital, Taipei 11217, Taiwan; thwu@vghtpe.gov.tw; 2Division of Allergy, Immunology & Rheumatology, Taipei Veterans General Hospital, Taipei 11217, Taiwan; darryliao@yahoo.com.tw (H.-T.L.); hctsai7@vghtpe.gov.tw (H.-C.T.); 3Division of Immunology & Rheumatology, Shin Kong Wu Ho Su Memorial Hospital, Taipei 11101, Taiwan; pearharry@yahoo.com.tw; 4Division of Transplantation Surgery, Taipei Veterans General Hospital, Taipei 11217, Taiwan; nclin@vghtpe.gov.tw (N.-C.L.); cychen6@vghtpe.gov.tw (C.-Y.C.); 5Institute of Molecular and Genomic Medicine, National Health Research Institutes, Zhunan 35053, Taiwan; pettsai@nhri.org.tw; 6Department of Urology, Taipei Veterans General Hospital, Taipei 11217, Taiwan; jayhuangx@gmail.com; 7Division of Allergy, Immunology & Rheumatology, Fu Jen Catholic University Hospital, New Taipei City 24352, Taiwan; 8Department of Internal Medicine, National Taiwan University Hospital, Taipei 100225, Taiwan

**Keywords:** B cell, immune repertoire (iR), complementary determining region 3 (CDR3), renal transplantation, clonal diversity, next-generation sequencing (NGS)

## Abstract

Background: Graft failure resulting from rejection or any other adverse event usually originates from an aberrant and/or exaggerated immune response and is often catastrophic in renal transplantation. So, it is essential to monitor patients’ immune status for detecting a rejection/graft failure early on. Methods: We monitored the sequence change of complementary determining region 3 (CDR3) in B-cell receptor (BCR) immunoglobulin heavy-chain (*IGH*) immune repertoire (iR) in 14 renal transplant patients using next-generation sequencing (NGS), correlating its diversity to various clinical events occurring after transplantation. *BCR-IGH-CDR3* in peripheral blood mononuclear cells was sequenced along the post-transplantation course by NGS using the iRweb server. Results: Datasets covering VDJ regions of *BCR-IGH-CDR3* indicated clonal diversity (D50) variations along the post-transplant course. Furthermore, principal component analysis showed the clustering of these sequence variations. A total of 544 shared sequences were identified before transplantation. D50 remained low in three patients receiving rituximab. Among them, one’s D50 resumed after 3 m, indicating graft tolerance. The D50 rapidly increased after grafting and decreased thereafter in four patients without rejection, decreased in two patients with T-cell-mediated rejection (TCMR) and exhibited a sharp down-sliding after 3 m in two patients receiving donations after cardiac death (DCD). In another two patients with TCMR, D50 was low just before individual episodes, but either became persistently low or returned to a plateau, depending on the failure or success of the immunosuppressive treatments. Shared CDR3 clonal expansions correlated to D50 changes. Agglomerative hierarchical clustering showed a commonly shared CDR3 sequence and at least two different clusters in five patients. Conclusions: Clonal diversity in *BCR**-IGH*-*CDR*3 varied depending on clinical courses of 14 renal transplant patients, including B-cell suppression therapy, TCMR, DCD, and graft tolerance. Adverse events on renal graft failure might lead to different clustering of *BCR* iR. However, these preliminary data need further verification in further studies for the possible applications of iR changes as genetic expression biomarkers or laboratory parameters to detect renal graft failure/rejection earlier.

## 1. Introduction

In renal transplantations, with the use of potent immunosuppressive agents immediately before and after grafting and in the maintenance phase thereafter, the incidence of acute rejection has dramatically fallen over time [1,2,3]. However, how an immune system adapts to the renal graft and maintains tolerance status under a set of complex immunosuppressants is still not fully understood [4,5].

An immune repertoire (iR) is the summation of T and B cells in a human body at any given moment [6,7,8,9,10,11,12]. It is both a snapshot and a historical record of an individual’s immune functions [13]. The application of iR sequencing on clinical medicine is mounting in recent years, such as those used for cancer characterization, psoriasis and human T-cell subset studies [14,15,16]. Presumably, sequencing the expressed T- or B-cell genes in the iR may have clinical implications in the prediction and clinical management for renal transplant rejection and there have been such reports [17,18,19]. A characterization of pre-transplant and post-transplant rejection risks by B-cell iR has revealed that patients who develop rejection have a more diverse iR before transplantation, suggesting a predisposing liability to rejection, and has also demonstrated a specific set of clonal expansion that would persist after the rejection [18]. Presumably, there might be a common pool of immunogenetic antigens that drive the rejection. Most sequence variation associated with immunoglobulins (Igs) and T-cell receptors (TCRs) are found in the complementary determining regions (CDRs), which are most variable [20]. CDR1 and CDR2 are found in the variable (V) region, but CDR3 includes some of V, all of diversity (D) region of heavy chains, and joining (J) regions [21]. Therefore, CDR3 is the most variable and can indicate more reliably the most timely immune-reaction events and immune status in transplant patients. In the present investigation, we analyze the CDR3 clone sequences in the immunoglobulin heavy chain (*IGH*) of the B-cell receptors (BCRs) in renal graft recipients under immunosuppressive treatments, and correlate these data to their clinical changes in post-transplant time. Although a similar investigation has previously been reported [19], we tried to apply a mathematical calculation mode to quantitatively analyze the immune diversity change. We are interested to observe if B-cell iR can provide alarms in advance for adverse clinical events, including rejection within the immune system.

Furthermore, we set up a next-generation sequencing (NGS) methodology that might potentially be used to evaluate the relationship between personal iR and outcome subtypes of or graft failure risks in renal transplant patients.

## 2. Patients and Methods

### 2.1. Patient Selection, Therapeutic Protocol, and Biochemical/Immunological Biomarker Detections

Fourteen closely monitored renal transplant patients were recruited for this study after signing an informed consent form. Their clinical backgrounds are listed in Table 1. All patients were subjected to a pre-transplant baseline and post-transplant follow-up iR study on CDR3 *IGH* of BCR and had baseline as well as follow-up blood sample collections for a total of 4 times (at the beginning (T0), 30 days (T1), 3 months (T2), and 12 months (T3) after transplant surgery). Patients #1 and #6 were transplant patients with blood-type mismatch. Patients #1 and #4 presented with a class I or class II panel-reactive antibodies (PRA-I or II), but not donor-specific antibodies (DSA), respectively [22,23]. Therefore, they (#1, #4, and #6) received preventive desensitization therapy with rituximab and intravenous immunoglobulin infusion (IVIG) before transplantation to deplete B cells for the prevention of severe and hyperacute rejection. Although Patient #8 also had a low titer of PRA-I, he was a blood-type-O compatible recipient with only two HLA mismatches. Therefore, he did not undergo premedication with rituximab. To further exclude the confounding effect exerted by rituximab, the T0 samples of these 3 patients were obtained before rituximab infusion. Major clinical features, including biochemical/immunological biomarkers and therapeutic treatment, as well as intervening clinical events of all 14 patients are provided in Table 2 and Appendix A. Standard biochemical/immunological protocol tests were performed in all patients at the individual blood-collection time points before the transplant surgery and along the post-transplant immunosuppressive treatments. These standard tests included blood urea nitrogen (BUN), creatinine (Creat), PRA-I, PRA-II, major histocompatibility (MHC) class-1 chain-related alloantibodies (MICAs) [24], CD3^+^, CD4^+^, CD8^+^, CD19^+^, CD^16+^CD^56+^ lymphocytes, and activated T cells (Appendix A) [25].

### 2.2. NGS for CDR3 of BCR IGH

The iR at pre- and post-transplant time points was evaluated by CDR3 sequencing of the BCR heavy chain (*IGH*). The whole experimental procedure included the careful recruitment of eligible patients, RNA extraction, iR library preparations, and MiSeq^®^ (Illumina, San Diego, CA, USA) sequencing.

#### 2.2.1. Blood-Collection Scheme

Three milliliters of peripheral venous blood samples were collected with 16 mm × 100 mm BD Vacutainer with ACD Solution A (#364606, 8.5 mL) Whole Blood Collection Tube (BD, Franklin Lakes, NJ, USA). Blood collection was performed on the day before transplantation (or before rituximab infusion in patients #1, #4, and #6), and 1, 3, as well as 12 month(s) after transplantation.

#### 2.2.2. RNA Extraction

RNA extraction procedures were conducted with Taigen LabPrep^®^ Total RNA Mini Kit (Taigen Biosciences, Taipei, Taiwan) from each 3.0 mL of freshly drawn peripheral venous blood in ACD solution A within 24 h after collection. The standardized RNA extraction protocol is described in the pamphlet provided by the manufacturer.

#### 2.2.3. iR Library Preparation Process

Two consecutive steps of reverse-transcription-polymerase chain reaction (RT-PCR) were performed with RNA as the starting material using proprietary Amplicon Rescued Multiplex (ARM) PCR technology with HBHI-M reagent system for human BCR *IGH* sequencing with Illumina MiSeq 300 PER, covering V and C genes (iRepertoire, Huntsville, AL; Han, Jian. Method for Evaluating and Comparing Immunorepertoires. Jian Han, assignee. Patent 9012148. 21 April 2015). One-step RT-PCR was performed to generate barcoded products using Qiagen One-Step RT-PCR Kit, Cat No. 210212 (Qiagen, Hilden, Germany). Multiplex PCR with Qiagen Multiplex PCR Kit, Cat No. 206143 (Qiagen) was performed using communal primers to generate sequencing templates. For better sequencing results, gel purification for size selection was performed to remove primer dimers using a Qiagen MinElute gel extraction kit, Cat No. 28604 (Qiagen). For BCR *IGH* –MiSeq compatible products, the gel band was arbitrarily set around 500 bp (490–570 bp), which was the size of the band extracted for the sequencing. A small portion of recovered libraries was electrophoresed to confirm its size. Size-selected library quantitation was performed by measurement using a Qubit^®^ fluorometer (Invitrogen, Carlsbad, CA, USA).

#### 2.2.4. MiSeq Sequencing

The sample concentration adjustment was based on Qubit^®^ fluorometer (Invitrogen) confirmation on library concentration and quantitative PCR (qPCR) using a 7900HT Fast-Real-Time PCR system. Pooling with the same amount of aliquot from each library was performed for sequencing template preparation. Ten aliquots of equal amounts were pooled together and mixed with Phix Control (Illumina, USA) in a ratio of 3:1; finally, 12.5 *p*M of product was loaded and sequenced with Illumina platform. Multiplex sequencing reaction was conducted with a MiSeq Reagent Kits v2 with 600 cycles and 2 × 300 bp output on a MiSeq sequencer (Illumina). The total output on reads *r*_1_, *r*_2, …,_
*r*_n_ fastq datasets was 10 Gb in an average run.

### 2.3. Data Analysis

Raw dataset fastq files were uploaded into the server maintained by iRepertoire for initial data processing. Data analyses were performed by iRweb software pipeline (iRepertoire, Huntsville, AL, USA) (https://irweb.irepertoire.com/nir/). D50 [27] was calculated by the following formula:

Assume that a sum (S) of samples include *r*_1_, *r*_2_, *r*_3_,……, and *r*_s_; *r*_1_ ≥ *r*_2_ ≥ *r*_3_ ……≥ *r*_i_ ≥ *r*_i+1_…… ≥ *r*_s_,

Define
(1)$$\sum_{i=1}^{S}ri=J$$
(2)$$\sum_{i=1}^{C}ri=C$$

If
(3)$$\sum_{i=1}^{C}ri>J/2$$

And
(4)$$\sum_{i=1}^{C-1}ri<J/2$$
D50 = 100*C*/*J*
where D50 is defined as 50% diversity of an iR of total number of CDR3s, including S distinct CDR3s in a ranked dominance configuration, where *r* stands for the amount (frequency) of individual CDR3; *r*_1_ is the amount of the most abundant CDR3; *r*_2_ is the amount of the second most abundant CDR3, and so on; *J* is the total number of distinct CDR3s; and *C* is the minimum number of distinct CDR3s amounting to ≥50% of total sequencing reads.

The Diversity Index (*DI*) is mathematically defined as follows:

Assume that the numbers (*n*) of unique CDR3 are:

*r*_1_, *r*_2_, *r*_3_,……, and *r_n_*

Where *r*_*i*_ is the amount (frequency) of the *i*-th CDR3 and *n* is the total number of unique CDR3s.
(5)$$\;\;\;\;\;\;\;\;\;\;\;\;DI=1-\overset{n}{\underset{i=1}{\sum }}{\left(\frac{rn}{r1+r2+\ldots +rn}\right)}^{2}$$

A calculation for Shannon entropy is presented for each sample [28,29]. The formula used in the calculation of the Shannon entropy (*H*) is:(6)$$H=-\sum_{i=1}^{10,000}pi$$

Only the top 10,000 CDR3s are included in this calculation, where *pi* is the amount (frequency) of *i*-th CDR3 within the top 10,000 CDR3s (in other words, 10,001th CDR3 and beyond are excluded in the calculation for *pi*).

### 2.4. Principal Component Analysis (PCA) and Agglomerative Hierarchical Clustering (AHC) for Pre-Transplant Baseline IGH iR 

PCA and AHC are carried out according to those previously reported [30,31,32]. A very convenient feature of the iRweb software is CDR3 algebra, which allows the comparison of the CDR3 sequences from one data set to another for identifying shared CDR3 profiles. This allows for a comparison among samples of different time points during the monitoring of disease fluctuation. All CDR3 frequencies were artificially scaled to 10 million reads to account for the differences in read depth among the samples, making comparisons between samples easier with this normalization step. A shared profile of baseline data in all renal transplant patients was obtained using this function. We constructed an observation/variable table based on this profile, using different samples as different observations and different CDR3 clone sequences as different variables. We then applied the PCA function of a commercial module of Excel for Microsoft Windows 10), namely XLSTAT (Addinsoft, USA) to calculate the PCA results on this set of baseline samples. Finally, the observation chart with PC1 and PC2 for dimension-reduced CDR3 listings of all samples was plotted for the evaluation of the observations based on the factor scores. The AHC function of XLSTAT was also applied to identify the clustering pattern of the same observations/variables table. A dendrogram showing the distinct clusters among the samples was sketched.

### 2.5. PCA and AHC for Post-Transplant IGH iR

Raw data in the form of CDR3 listings from the serial detection of each patient were processed with the same pipeline of iRweb and analyzed using the PCA and AHC functions of XLSTAT module in a PC windows environment. All transplant patients undergoing immunosuppressive treatment were processed.

### 2.6. Statistical Analysis

Data were compared with the chi-squared test, or Student’s paired *t*-test where appropriate. *p*-value < 0.05 was considered significantly different between the groups.

## 3. Results

### 3.1. CDR3 Sequence, D50 and DI Analyses Show Distinct Profiles of Changes in iR

The baseline demographic and clinical data of 14 renal transplant patients are shown in Table 1, Table 2 and Appendix A. Patients #1 and #6 had blood type mismatch. Patients #1 and #4 had PRA-I or II, but not DSA. Therefore, we administered preventive desensitization therapy with rituximab and IVIG before transplantation to deplete B cells in these 3 patients. Patient #8 presented with a low PRA-I titer, but had an O blood type, compatible with the donor, and only had 2 HLA mismatches. Therefore, he did not undergo rituximab. There was only one patient (#13) developing a low titer of de novo DSA at T3, which seemed to not elicit any significant clinical adverse symptoms. Before transplantation, the values of D50 at T0 in patients #1, #2, #3, #4, #5, #6, #12, and #13 were below 10, but only the values at T1 in patients # 1, #4, and 6# (all of whom were treated with rituximab) were below 10, as shown in Table 3 (*p* = 0.053 as calculated by the chi-squared test), suggesting the low diversity of B cells in the samples before transplantation and higher diversity of B cells in those immediately after transplantation. In fact, there were 1,000,751 CDR3 and 89,712 unique CDR3 reads on average, detected for all samples with a mean D50 value of 11.18 ± 10.12 at T0 and 21.04 ± 14.31 at T1 (*p* = 0.016 as calculated by Student’s paired *t*-test); DI of 20.20 ± 11.49 at T0 and 27.53 ± 14.86 at T1 (*p* = 0.046 as calculated by Student’s paired *t*-test); and H of 11.31 ± 1.60 at T0 and 11.04 ± 3.50 at T1 (*p* = 0.706 as calculated by Student’s paired *t*-test). DI and H are alternative measurement values of diversity. As shown in the above, “DI” can also be alternatively defined as 100 minus the area under the curve between the percentage of total reads and percentage of unique CDR3s, when the frequencies of unique CDR3s are accumulated from most frequent to least frequent. “Entropy (H)” is defined as that of the Shannon entropy [28,29]. There was no single whole BCR *IGH* CDR3 sequence expressed in all 14 patients. However, there was a profile of nucleotides containing 544 *IGH* CDR3 sequences expressed across some of the patients, as listed in Appendix A. “ARDLDY” is this shared CDR3 sequence being expressed in 7 out of 14 baseline samples.

For the studies after the transplantation, the results are presented as serial detections of BCR *IGH* iR, i.e., datasets obtained from all 14 patients, with each having pre-transplant and post-transplant blood collection. The average D50 after transplantations was 21.04 ± 14.31, which was significantly higher than that (11.18 ± 10.12, *p* < 0.05 as calculated by Student’s paired *t*-test) before transplantations. The average D50 values from all 14 patients at all times were 18.0, and there were 925,113 CDR3 reads (86,012 unique CDR3) detected for each sample (Table 3). Three categories of D50 trends were observed. Among them, D50 remained low after transplantations for a prolonged time, suggesting low clonal diversity in the long post-transplant period for three patients who received rituximab and intravenous immunoglobulin (desensitization) treatment, i.e., patients #1, #4, and #6 (Figure 1a). However, D50/DI returned to a much higher level, soon after the renal graft became stable (after T2) only in patient #4 who suffered from a short-term *E. coli* urinary tract infection at one month (Table 2 and Figure 1a). The courses of patients #1 and #6 were much more ominous, which might be caused by too much immunosuppression (two doses of rituximab as shown in Table 2), resulting in polyoma viral infection (at 4.5 months in patient #1 and 2 months in patient #6, with (at 4 months in patient #6) or without polyoma nephropathy (designated at T2 and T3 in Table 2), ABO incompatibility, and presence of PRA as well as high HLA mismatch (Table 1). In addition, patient #6 also suffered from T-cell-mediated rejection (TCMR) occurring 6 months after transplantation. D50 and DI values exhibited abrupt elevation immediately after transplantation, and showed only a subtle decline thereafter in 4 patients (patients #8, #10, #11, and #13) who either had transient BK viral infection or urinary tract infection (UTI), but did not have any rejection episode (Figure 1b). Patients #5, #6, #7, #9, #12, and #14 suffered from TCMR, which was parallel to the decline in D50/DI (Figure 1a–c). Patients #12 and #14 seemed to exhibit an increase in D50/DI after the episodes, which was possibly because of rapid and effective therapies to ameliorate the rejections. In fact, patient #12 suffered from a lymph leakage (data not shown) in the abdomen, which was treated with embolization, and an independent episode of UTI with *E. coli*, together with the development of TCMR. Patients 2# and 3# received donations after cardiac death (DCD), and so inevitably developed delayed graft function (DGF) (Table 2 and Figure 1b). As shown in Table 2 and Figure 1b, they both suffered from a high Perico score and a sharp decline in D50 more than 1 month (T1) after transplantation. It is unclear whether this decline is relevant to the high Perico score, and a prolonged ischemia time resulted from a collapse of the donor’s circulation, which led to the subsequent DGF. On the other hand, 3 patients (patients #7, #9, and #14), free of rituximab treatment, suffered from TCMR episodes, and exhibited either a decline in (#7 and #9) or an elevation of the values (#14) in D50/DI (Figure 1c). Patient #14, although suffering from TCMR, was successfully treated and exhibited a return of the D50 and DI. An event biopsy showed equivocal microvascular inflammation (Table 2), but there was no de novo development of PRA or DSA. Therefore, there was no sufficient evidence to support a diagnosis of antibody-mediated rejection (AMR). It is worth noting that entropy (11.31 ± 1.60 at T0 and 11.04 ± 3.50 at T1, *p* = 0.706, which was the most remarkable difference presented by the entropies) remained irrelevant to the disease status of these patients without conspicuous fluctuation in all patients throughout the whole course of the observations, suggesting it was not likely to be a sensitive biomarker. The serum creatinine levels of these patients at different times were correlated to the adverse events, as well as fluctuations of D50/DI/H. Although the creatinine levels were well correlated to the adverse events, the times of D50/DI/H changes seemed to occur before the creatinine change, although they were not statistically verified. This implies that personal iR indices may be also regarded as good indicators for post-transplant monitoring of immune status.

### 3.2. Distinct Clusters as Shown by PCA and AHC in Pre-Transplant Baseline IGH iR

Apart from the shared CDR3 analysis, we also performed PCA on the 14 pre-transplant baseline datasets and tried to analyze numerical data (CDR3 listings) structured in a 14 observations/544 variables table, as previously described. We analyzed the CDR3 sequence listings as principal components 1 and 2 (PC1 and PC2) for a 2D visualization. The PC1 and PC2 of CDR3 listings from 14 pre-transplant samples (T0) are presented as Figure 2a. We found that the CDR3 listings from pre-transplant samples held a convergent profile, except that of samples 12 (patient #8), 13 (patient #9), 14 (patient #10), 15 (patient #11), and #19 (patient #14). AHC analysis suggested that there were several (at least 2) clusters that were formed (Figure 2b,c). The cause of absence of convergence was unknown, but it might indicate that the original pathogenic mechanisms underlying the ESRDs of these five patients were quite different from those of the other patients. Furthermore, the three convergent clusters might indicate that the other remaining eight patients might have experienced at least two different pathways to result in ESRD.

### 3.3. Distinct Clusters as Shown by PCA and AHC in Post-Transplant IGH iR

The PCA results of *IGH* datasets on serial detections of all patients after transplantation were plotted as principal components 1 and 2 (PC1 and PC2) for a 2D visualization (Figure 3a). PC1 and PC2 data points within the same patient exhibit a convergent pattern for all patients, except that for samples 3 (patient #3), 11 (patient #7), 17 (patient #12), and 18 (patient #13). Patient #3 exhibited a slight consistent horizontal shift along the PC1 axis, whereas patients #7, #12, and #13 showed a slight consistent vertical shift along the PC2 axis. Interestingly, two of them (patients #7 and #12) exhibited a final fall-down of D50/DI at the end of the study. Both of them suffered from TCMR. The correlation between the PCA pattern with respect to the clinical phenotype in patient #13 has yet to be examined with more clinical biomarkers. AHC analysis confirmed that there were several (at least two) clusters formed (Figure 3b,c), indicating that there might have been several (at least two) shared mechanisms to help maintain graft tolerance.

## 4. Discussion

### 4.1. D50 Profile Trajectory as a Good Companion Indicator for Predicting Graft Dysfunction/Failure and/or Rejection Episodes

There have been a few studies dealing with the iR of T cells in human kidney transplantation [17,33,34]. However, not many studies have been dedicated to the iR of B cells in the same population, especially in patients undergoing immunosuppressant treatments [21]. The present investigation might have demonstrated, for the first time, a sequential change in the iR of BCR in renal transplant patients after grafting, who were undergoing immunosuppressant treatments. It is accepted that clonal diversity generally decreases when a host confronts diseases or inflammation [35,36]. However, in patients receiving rituximab treatment, D50 was extremely low, indicating that CDR3 clonal diversity responds to the CD20 monoclonal antibody, causing B-cell clonal depletion, which has been well documented [37]. The CDR3 region is particularly unique as the antigen specificity is highly correlated to this region of the BCR. The D50, which is a quantitative measure of the degree of diversity of B cells within a sample, is the percentage of dominant and unique B-cell clones that account for the cumulative 50% of the total CDR3s measured in the sample. The more diverse a library, the closer the value is to 50. Low D50 is associated with a decreased diversity. BCR *IGH* datasets and CDR3 count listing in these 14 patients reflect the changes in immune status along the whole renal transplantation course (Figure 1 and Table 2). The three categories of outcome trends could also be demonstrated using DI and H. However, entropy H did not show a close association with the adverse or favorable events in most of the patients throughout a lengthy period of the observation time, suggesting it might not be used as a reliable and sensitive marker to monitor the changes in iR. On the other hand, we also tried to correlate the fluctuation of serum creatinine levels to the changes in diversity in iR. Indeed, the serum creatinine levels or clinical renal function did show a significant relationship with the adverse events or graft failure, and iR change seemed to occur before the events and creatinine changes, which usually occurred after renal functions were substantially disturbed (data not shown). This is probably because the iR change is the driver of the body’s defense against extrinsic offending events, and serum creatinine change is the outcome of the body’s defense mechanism. Therefore, the level of creatinine was less likely to exhibit conspicuous fluctuations in such a short time of monitoring (e.g., one or two weeks in the case of chronic rejection). Alternatively, the iR changes may be a harbinger of favorable or adverse events, if more data are accumulated. 

In monitoring the post-transplant courses, we identified three categories of changes in iRs, as shown in Figure 1a–c, which were related to rejection episodes, graft failures, or other unfavorable events. It is conceivable that they correlated well to immunosuppressant treatments, and/or polyoma viral infection events, rejection episodes, but not intervening bacterial infections, including UTIs. In fact, as demonstrated in Figure 1b, infections seemed to exert a negligible effect on the IR of BCR in patients #8, #10, #11, and #13. In the rituximab treatment (desensitization) group, including patients #1, #4, and #6 (Figure 1a), the D50 and DI exhibited a delayed return to height (i.e., normalized) level because of the powerful and sustaining cytotoxic effect of the drug on the B cells. This implies that close and patient observations are essential for monitoring renal transplant patients who are treated with B-cell-depleting agents. Among the three patients, patient #4 experienced a much smooth course, with only one episode of infection rapidly cured by antibiotic administration. The other two patient experiences were complicated by infection and polyoma viral nephropathy of the graft, which led to delayed recovery of the *BCR* IR. Long term follow-up of these two patients revealed that they were eventually normalized (data not shown). Thus, a recovery of the clonal diversity might occur one year well after the administration of rituximab. In the group with a rapid elevation of D50 values after transplantation and a steadily gradual downward trajectory thereafter (patients #8, #10, #11, and #13 in Figure 1b), we did not detect any rejection episodes. Only patient #13 developed a very mild DSA at the last time of blood sampling, which could be reflected by a tiny dip in D50/DI at T3, as demonstrated in Figure 1b. This indicates that D50 can reliably reveal the stable tolerance of renal graft. A delayed modest decline in D50 is inevitable because a variable tendency of chronic rejection or delayed renal graft failure is common in long-standing tolerant renal recipients. On the other hand, patients #2, #3, #5, and #12 (Figure 1b) exhibited a much more ominous course. Patients #2 and #3 were recipients of donations after cardiac death. It was unclear if this factor could have any effect on graft survival. However, we did observe a sharp decline in D50 sometime after transplantation in both patients, although patient #2 was later rescued by intensifying immunosuppressants. Patients #5 and #12 received a TCMR approximately 3 months (T2) after the transplantation, parallel to the decline in D50. Intriguingly, the abrupt decline in D50/DI was apparently slightly earlier than the actual event (Figure 2b and Table 2), indicating that declined D50/DI may be regarded as a harbinger for rejection. Patient #12 experienced a bumpy post-transplant course, with the development of UTI, lymph leakage, and two episodes of TCMR events. Indeed, except for the last episode of TCMR, these episodes did not preclude the return of D50 to a high (normal) level because they were individually controlled by appropriate treatments. In the group that did not undergo rituximab therapy, but still experienced an initial decline in D50/DI (Figure 1c), all 3 patients suffered from TCMR, which occurred after D50/DI showed a decreasing trend. However, patients #9 and #14 only experienced a mild disease because they were quickly rescued, even without the appearance of a decline in D50/DI, as indicated in Figure 1c. It was possible that TCMR might have been initiated just after the transplantation surgery, because D50/DI never followed an upward trajectory from the very beginning of immunosuppressive therapy. This may suggest that, if D50/DI does not increase at the 1st measurement after transplantation, vigorous efforts should be made to find a primary graft failure or rejection. On the contrary, this may also suggest that our current NGS protocol for iR is still too insensitive to detect early signs of graft rejection, or a more frequent measurement is mandatory to find it. On the other hand, although patient #14 had a polyoma viral infection, the course of recovery of D50/DI was smooth, similar to those observed in patients #1, #6, #10, and #11, implying that BK viral infection or polyoma nephropathy, if detected early on and well treated, does not have a conspicuously harmful effect on renal graft tolerance. It is worth noting that this particular patient (#14) was found to have microvascular inflammation in a biopsy event, but since she did not have de novo PRA or DSA development, AMR was excluded. Otherwise, there was no patient that developed AMR in our cohort.

Thus, in general, a patient who has successful renal transplantation and an uneventful post-transplant course is expected to have a rapid increase in immune diversity (D50 at T0 vs. D50 at T1, *p* = 0.053 by chi-squared test; if excluding 3 patients with rituximab therapy, all patients exhibited an immediate return of D50) after transplant surgery, but may experience a very gradual decrease in immune diversity thereafter because of intercurrent infections, transient episodes of mild TCMR, or other unidentified harmful events. The present study had some shortcomings. First, because of the tedious procedures used to analyze IR diversity and the cost, it was impossible to frequently measure iR throughout the whole course of the post-transplant immunotherapy. Second, as mentioned above, undetectable intervening infections and/or graft dysfunction may occur and were corrected so rapidly by the host immune system that changes in D50 escaped the detection time point for iR analysis. Further work should be conducted to spare the cost and simplify the procedures of NGS analysis in renal transplantation. In conclusion, the D50 profile trajectory might potentially be used as a companion indicator for predicting graft-rejection episodes or failures, but cannot be used to detect other adverse events, such as bacterial infection/polyoma events. Indeed, our results may be superior to those previously reported as methods for monitoring immune status in renal transplant patients [26].

### 4.2. Clusters of Changes in Immune Diversity as a Good Indicator for Searching Renal Failure Etiologies and Specific Gene Finders during Adverse Events in the Course of Renal Transplantation

In terms of PCA presentations in these datasets, the baseline values (T0) from pre-transplant samples did have a clustering effect (Figure 2), implying that there is a specific clonal expansion pattern for specific diseases. We performed a shared profile analysis in patient datasets of disease baselines to identify shared dominant CDR3 sequences. A shared profile of 544 CDR3 clone sequences were identified, i.e., clone-expansion sequences were present in at least two patients. Sequence “ARDLDY” (representing alanine–arginine–aspartate–leucine–aspartate–tyrosine at peptide level) is the most significant CDR3 expansion sequence, which was present in 7 out of 14 patients. The significance of this finding needs further in-depth investigation.

A PCA analysis of this shared profile showed a convergent pattern, except for samples #12, 13, 14, 15, and 19 (patients # 8, #9, #10, #11, and #14). The variable clinical phenotypes of these five patients may be contributed to by additional confounding factors, which would be clearer if investigated further.

On the other hand, we also performed shared profile analysis on post-transplant datasets (T1, T2, and T3) to identify shared dominant CDR3 sequences in patients undergoing immunosuppressant treatments. The PCA on the CDR3 shared profile showed a divergent pattern in samples #3, #11, #17, and #18 (patients #3, #7, #12, and #13), indicating changes in immune diversity. Patient #3, who was a recipient of DCD, received a stable course without any bacterial or polyoma viral events. Patients #7 and #13 had suspected infection episodes, but without rejection. Their relationship with clinical phenotypes deserves further clarification. Taken together, the changes in immune diversity as measured by PCA divergent profiles can be a potential indicator for follow-up investigations on clinical phenotypes. It might have the potential to spare the risky and invasive protocol biopsy procedures [38,39].

The CDR3 shared profiles of serial detections also had a clustering effect for individual patients, as illustrated in those of patients #3, #7, #12, and #13, supporting our hypothesis that specific clonal expansion patterns in an individual patient may occur in response to some unique exogenous stimuli, such as post-transplant immunosuppression treatment, regardless of grafting success.

## 5. Conclusions

We may have successfully established a pipeline of experimental design, data acquisition, processing, and analysis to be used in monitoring the changes in immune diversity as related to the outcomes of various kidney transplantations. The experimental procedures include the timely collection of whole blood samples, immediate RNA extraction, sequencing library preparation, and NGS to obtain BCR CDR3 iR profiles, which were then analyzed by PCA for downgrading complex dimensions to make sense of CDR3 sequence variations. The BCR CDR3 sequences exhibited a clustering pattern as represented by PCA for individual patients undergoing serial detections, suggesting a clonal expansion specific for individual patients in their individual scenarios. In addition, a shared profile of CDR3 sequences was demonstrated to help identify shared clonal expansion, which, in turn, could be used to complete a corresponding PCA and AHC. These two analyzing procedures can be used to correlate the changes in immune diversity (D50 or DI, but not entropy) in renal transplant patients undergoing immunosuppressive therapies. The immune diversity is closely related to graft failure/rejection, but not bacterial infection/polyoma events. However, with the attainment of these limited data, we could not attribute any diversity changes to a specific event. For the purpose of this verification, it is necessary to collect a large number of samples.

There were some quite inevitable defects that occurred in the present investigation, i.e., it was difficult to recruit the patients because of the low incidence of renal transplantations and low consent rate of enrollments in this country, as well as higher costs of individual iR sequencing. The small number of enrollments directly resulted in the heterogeneity of stratified groups. Moreover, V-gene usage and class-switching of the immunoglobulins were not explored. Although the experimental procedures were quite preliminary and complex, we propose that further investigations in this field may help to establish a convenient and reliable immunogenetic biomarker to be used as a good tool for following up the clinical course of individual renal transplant patients. The way to interpret various responses in individual renal transplant patients using the iR information deserves further investigation with a larger sample size.

## Figures and Tables

**Figure 1 jcm-11-02980-f001:**
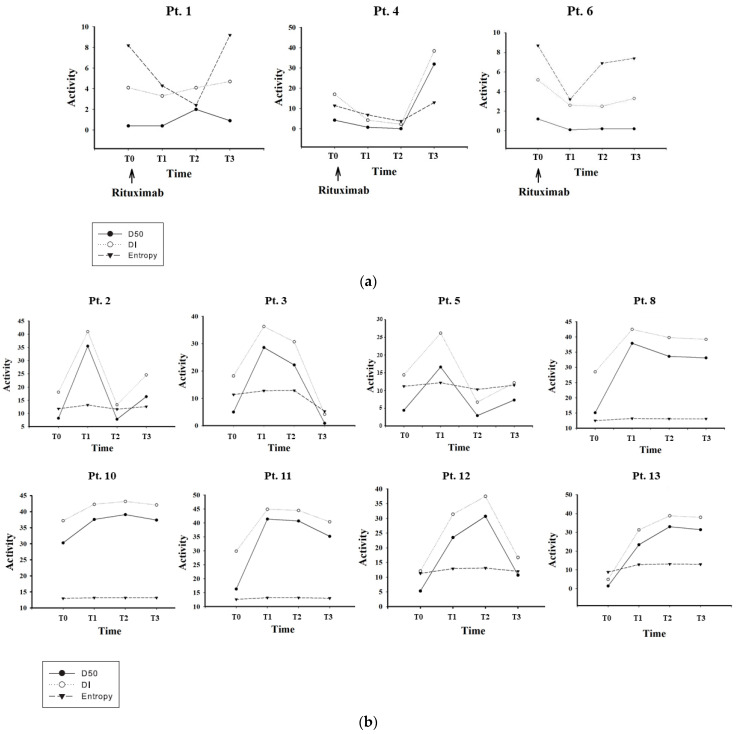
The fluctuations in DI, D50, and entropy (H) along the course of renal transplantation, including the post-transplant immunosuppressive treatments. (**a**) Patients #1, #4, and #6 who underwent rituximab therapy for desensitization before transplantation because of predicted high risks of rejections; (**b**) patients #2, #3, #5, #8, #10, #11, #12, and #13 who exhibited a rapid elevation of D50 after transplantation; (**c**) patients #7, #9, and #14 who had a primary reduction in D50 after transplantation, and all of those who developed T-cell-mediated rejection. All the 1st samples were obtained just before transplantation surgery (T0), except those in desensitized patients (#1, #4, and #6), whose T0 was just before the 1st dose of rituximab infusion. Subsequent sampling times were 30 (T1), 90 (T2), and 360 days (T3) after the transplantation. Patient #9 died from suicide just before T3 because of psychological problems; Pt: patient.

**Figure 2 jcm-11-02980-f002:**
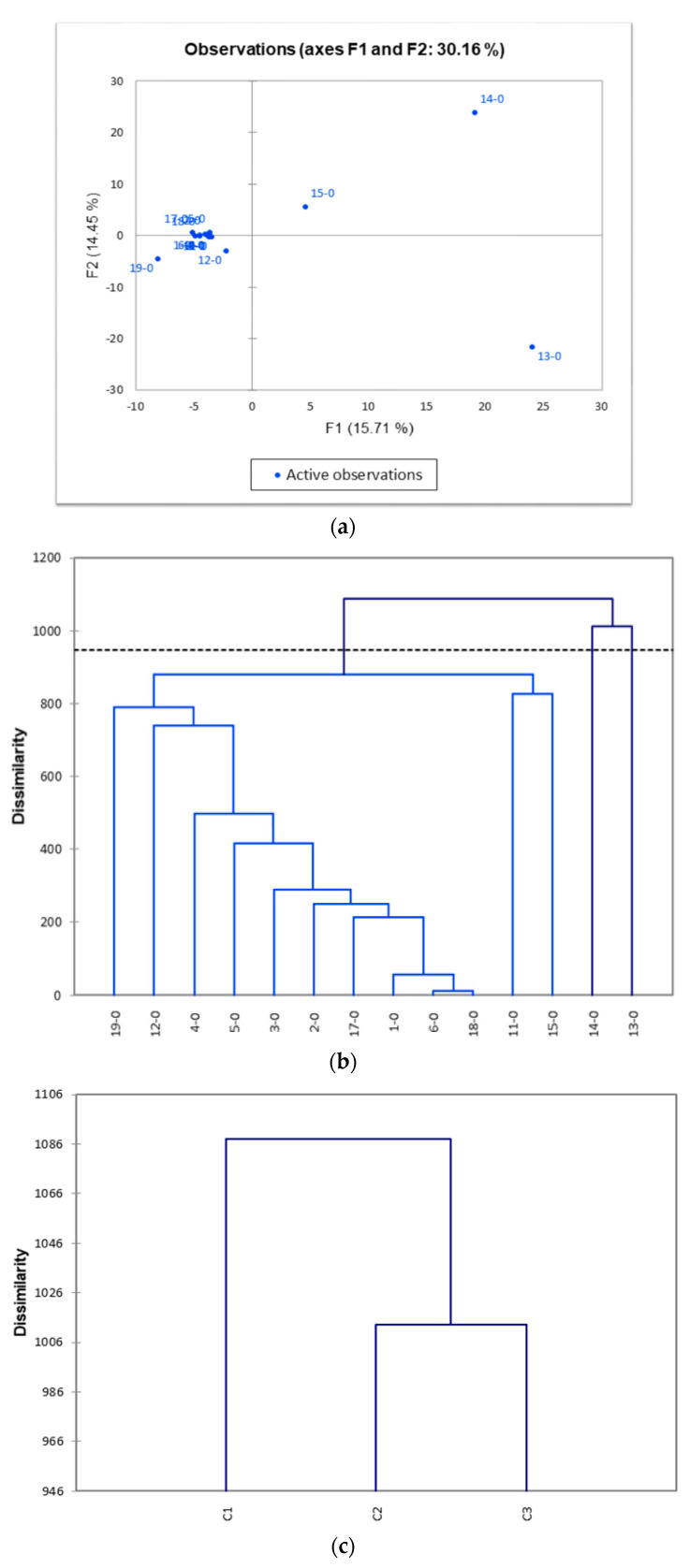
Principal component analysis (PCA) and agglomerative hierarchical clustering (AHC) for the pre-transplant baseline *IGH* repertoires in all 14 renal transplant patients. (**a**) The PCA charts represent the observations in the PCA space. A new set of variables or principal components (PCs) were created, explaining the maximum variance of datasets, followed by PC2, and so on. X and Y axes show principal components 1 (F1) and 2 (F2), and the percent variation explained by each component is shown in parenthesis; (**b**) sample view and (**c**) cluster view of AHC, C1, C2, and C3 on X axis represent different clusters of gene expression. All the numbers in the figures represent sample numbers rather than patient numbers due to original designations. the numbers 1-0, 2-0, 3-0, 4-0, 5-0, 6-0, 11-0, 12-0, 13-6, 14-0, 15-0, 17-0, 18-0, and 19-0 stand for samples from patient number 1, 2, 3, 4, 5, 6, 7, 8, 9, 10, 11, 12, 13, and 14 at T0 (just at the time of transplantation surgery).

**Figure 3 jcm-11-02980-f003:**
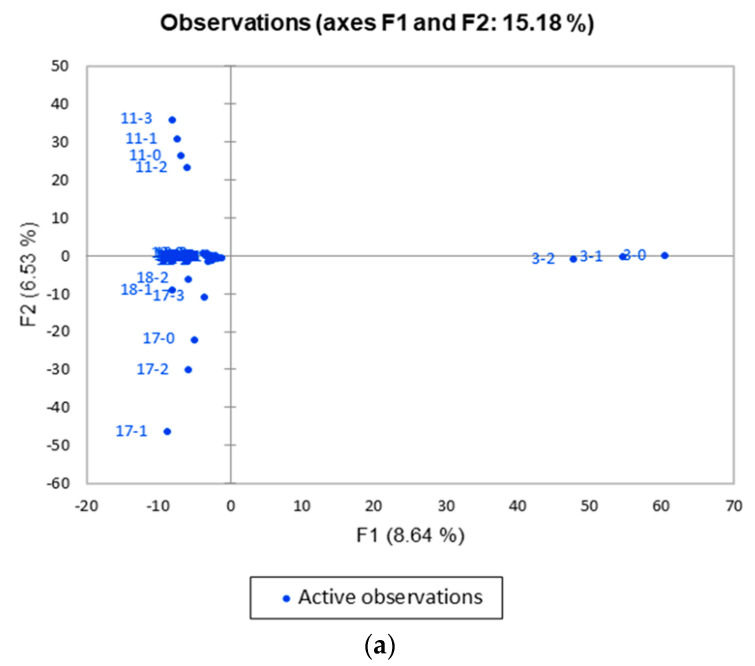
Longitudinal principal component analysis (PCA) and agglomerative hierarchical clustering (AHC) to monitor *IGH* repertoires in all 14 renal transplant patients. (**a**) The PCA charts represent the observations in the PCA space. A new set of variables or principal components (PCs) were created to describe more complex variability in the datasets. The first PC (PC1) explains the maximum variance of the datasets, followed by PC2, and so on. X and Y axes show principal components 1 (F1) and 2 (F2), and the percent variation explained by each component is shown in parenthesis. Time sequence of blood sample drawn is marked as separate sample numbers in an ascending order. (**b**) Longitudinal AHC for *IGH* repertoires in the patients (sample view) and (**c**) cluster view, C in X axis represents individual cluster of gene expression. All the numbers in the figures represent sample numbers rather than patient numbers due to original designations. (The numbers 1, 2, 3, 4, 5, 6, 11, 12, 13, 14, 15, 17, 18, and 19 represent the samples from patient numbers 1, 2, 3, 4, 5, 6, 7, 8, 9, 10, 11, 12, 13, and 14, whereas the 0, 1, 2, and 3 following the hyphen represent the times T0, T1, T2, and T3 when the blood was sampled).

**Table 1 jcm-11-02980-t001:** Basic demographic features of 14 renal transplant patients.

	Recipient Age/Sex	Donor Age/Sex	Recipient ABO/Rh	Donor ABO/Rh	Allograft Source	HLA Mismatch	Cause of ESRD	Perico Score *	PRA-I *	PRA-II *	Remark
1	64/F	62/M	O/+	A	L	3	?	<3	56.3	N	DS
2	46/M	51/M	AB/+	AB	C	2	CIN	6	N	N	DCD
3	65/F	51M	AB/+	AB	C	2	CGN	6	N	N	DCD
4	63/F	67/M	A/+	O/+	L	5	DMN	3	N	17.8	DS
5	55/M	24/M	B/+	B/+	L	3	?	<3	N	N	
6	34/F	60/M	B/+	AB/+	L	3	IgAN	3	N	N	DS
7	40/M	61/F	A/+	A/+	L	2	DMN	<3	N	N	TCMR
8	39/M	61/F	O/+	O/+	L	2	Nscl	<3	8.7	N	
9	49/F	69/F	O/+	O/+	L	3	?	<3	N	N	TCMR
10	38/F	43/F	A/+	A/+	L	5	DMN	<3	N	N	
11	59/F	63/M	O/+	O/+	L	3	?	<3	N	N	
12	63/M	59/M	B/+	O/+	L	6	CIN	3	N	N	
13	36/F	70/M	O/+	O/+	L	2	LN	3	N	N	
14	40/F	52/M	A/+	A/+	C	2	Nscl	3	N	N	TCMR

F: female, M: male; ABO/Rh: blood type A, B, or O and Rhesus; HLA: human leukocyte antigen; ESRD: end-stage renal disease; PRA: panel reactive antibodies, I: class 1 major histocompatibility, II: class 2 major histocompatibility; L: living graft; C: cadaver graft; ?: the etiology was uncertain; CIN: chronic interstitial nephritis; CGN: chronic glomerulonephritis; IgAN: IgA nephropathy; DMN: diabetic nephropathy; Nscl: nephrosclerosis; LN: lupus nephritis; Perico score: Assessment of the quality of renal graft [26]; * measured at the time zero (before transplantation). DS: desensitization; DCD: donation after cardiac death; TCMR: T-cell-mediated rejection.

**Table 2 jcm-11-02980-t002:** The clinical events of all 14 renal transplant patients along the course of immunosuppressive therapy.

Patient	DGF	Infection			TCMR			AMR			Desensitization *
		T1	T2	T3	T1	T2	T3	T1	T2	T3	
**#1**				BKV, 4.5 M							−21 D, −1 D
			UTI (*Ec*), 3 M	UTI, 12 M							
**#2**	+										
**#3**	+										
**#4**		UTI (*Ec*), 1 M									−21 D
**#5**			UTI (*Ef*), 1.5 M								
**#6**			BKV, 2 M	PVN, 4 M							−21 D, 0 D
**#7**			PA, 1 M	UTI (*Pm*, *Ec*), 9 M							
**#8**	+	CVP (*Se*), 7 D									
**#9**											
**#10**				BKV, 8 M–10 M							
**#11**				BKV, 12 M							
**#12**		UTI (*Ec*), 1 M			1 M						
**#13**		UTI (*Kp*), 9 D	UTI (*Kp*), 2 M								
**#14**	+	AWA, 1 M	CMV, 2 M	PVN, 4.5 M	20 D	4 M	9 M			MVI (?) 9 M	

DGF: delayed graft function; TCMR: T-cell-mediated rejection; AMR; antibody-mediated rejection; CMV: cytomegaloviremia; BKV: polyoma virus infection; PA: peritoneal abscess; PVN: polyoma virus nephropathy; CVP: central venous catheter colonization; MVI (?): microvascular inflammation, which was only equivocally present but AMR was not supported because there was no de novo donor-specific antibodies; *Pm*: *Proteus mirabilis*; *Ec*: *Escherichia coli*; *Kp*: *Klebsiella pneumoniae, Ef: Enterococcus fecalis; Se: Staphylloccus epidermidis*; UTI: urinary tract infection; AWA: abdominal wall abscess; * Desensitization includes rituximab and double filtration plasmapheresis followed by intravenous immunoglobulin injection to prevent immediate rejection after transplantation with the second dose (as induction dose if necessary) only with rituximab. M: month(s); D: day(s); −21 D, −1 D, and 0 D = 21 days or one day before transplantation and just on the day of transplantation. Column T1 represents the period between T0 and T1, T2 represents period between T1 and T2, and so on. The individual bacterial infections were verified by culture results; CMV infection was confirmed by polymerase chain reaction; TCMR, AMR, or PVN was confirmed by histopathology; and BKV was confirmed by decoy cells in urine cytology.

**Table 3 jcm-11-02980-t003:** Longitudinal follow-up of pre- and post-transplantation changes in CDR3 sequence expression, as well as their immune diversity fluctuation.

Sample	CDR3 Reads	Unique CDR3 Reads	D50	DI	H
1-T0	332,605	8707	0.4	4.1	8.2
1-T1	44,829	906	0.4	3.3	4.3
1-T2	2709	102	2	4.1	2.4
1-T3	342,511	12,875	0.9	4.7	9.2
2-T0	1,350,497	114,401	8.2	18.1	11.8
2-T1	1,085,401	100,641	35.5	41	13.2
2-T2	201,291	15,366	7.8	13.3	11.6
2-T3	660,271	43,697	16.4	24.6	12.6
3-T0	1,153,784	137,153	5	18.2	11.4
3-T1	1,179,141	122,205	28.9	36.3	12.8
3-T2	841,419	58,891	22.2	30.7	12.9
3-T3	25,797	787	0.9	4.2	5.3
4-T0	1,137,969	92,997	4.2	17	11.4
4-T1	94,964	3075	0.7	4.2	6.8
4-T2	448,670	3028	0	2.2	3.7
4-T3	1,005,018	114,697	31.9	38.4	12.9
5-T0	1,402,168	84,251	4.4	14.4	11.2
5-T1	627,283	42,274	16.6	26.2	12.2
5-T2	294,124	14,043	2.9	6.7	10.3
5-T3	734,429	32,048	7.3	12.2	11.5
6-T0	151,185	6366	1.2	5.2	8.7
6-T1	71,825	831	0.1	2.6	3.2
6-T2	948,704	9602	0.2	2.5	6.9
6-T3	950,614	17,209	0.2	3.3	7.4
7-T0	1,183,078	132,636	23.9	33.4	12.6
7-T1	1,287,151	105,115	21.1	31.2	12.3
7-T2	1,207,732	120,182	22.4	32.5	12.6
7-T3	652,595	68,099	18.5	29.7	12.4
8-T0	1,645,902	168,487	15.1	28.5	12.5
8-T1	1,140,959	189,932	37.9	42.5	13.2
8-T2	1,258,039	157,665	33.6	39.8	13.1
8-T3	845,907	89,271	33.1	39.2	13.1
9-T0	1,313,895	144,136	27.6	35.6	12.7
9-T1	1,112,684	171,721	14.8	28.9	12.2
9-T2	1,047,150	70,334	21.2	30.3	12.6
10-T0	1,061,414	118,472	30.3	37.2	13
10-T1	918,080	200,001	37.6	42.3	13.2
10-T2	1,372,288	199,626	39.1	43.2	13.2
10-T3	906,739	105,136	37.4	42.1	13.2
11-T0	1,163,277	129,036	16.3	29.9	12.6
11-T1	1,361,199	233,772	41.4	44.9	13.2
11-T2	1,124,920	182,482	40.7	44.5	13.2
11-T3	1,123,497	183,386	35.2	40.4	13
12-T0	866,775	43,874	5.3	12.1	11.3
12-T1	2,220,129	111,706	23.5	31.4	12.9
12-T2	2,449,184	143,081	30.7	37.5	13.1
12-T3	779,336	39,210	10.7	16.7	12
13-T0	193,485	7240	1.4	4.9	8.8
13-T1	883,178	64,490	23.4	31.3	12.8
13-T2	979,861	83,809	33	38.8	13.1
13-T3	1,448,968	130,378	31.4	38	12.9
14-T0	1,054,484	68,213	13.2	24.2	12.1
14-T1	926,726	54,854	12.6	19.3	12.2
14-T2	1,050,148	80,138	27.9	34.3	13
14-T3	1,215,236	98,127	32.9	38.8	13
Average	925,113	86,012	18.0	24.7	11.1

Sample number is represented as patient No.–time point of sampling, i.e., 1-T1 means patient #1 sampled at day 30 (1 month). D50 = 50% diversity of iR; DI = diversity index; H = entropy.

## Data Availability

Data supporting reported results can be found at https://irweb.irepertoire.com/nir/.

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
