# Peer review of "High-Throughput Sequencing of Complementarity Determining Region 3 in the Heavy Chain of B-Cell Receptor in Renal Transplant Recipients: A Preliminary Report"

_jcm, 2022, doi:10.3390/jcm11112980_

Round 1

Reviewer 1 Report

Minor Comments

IS CDR3 just the total read count in the figures? Unclear how this is helpful? Is there a way this is used to normalize the data, what is the significance of this value?

Conflicting statements. ‘There was no single BCR IGH CDR3 sequence being exhibited in all 14 samples. However, there was a shared profile?  Can you please complain what the shared profile means? Define how to get something shared, when none of the sequences are the same.

There are still English grammar mistakes throughout the manuscript.

The clinical phenotype in patients 13 has yet to be examined.  Why not? This would help increase the ability to understand what is happening in this patient since the sample size here is very small. Are the data unavailable? Should this patient be excluded from the study for clarity of the research?

Although the authors provide some explanation of why serum creatinine may not correlate with iR, more explanation is needed.  Since creatinine is a direct measure of kidney function, the explanation suggests that there is no relevance of iR in patients. 

Author Response

Reviewer 1

IS CDR3 just the total read count in the figures? Unclear how this is helpful? Is there a way this is used to normalize the data, what is the significance of this value?

Ans: Thank you for your comments. Since the antigen receptors are composed of  variable domains, heavy and light chainand most sequence variation associated with Ig’s and T cell receptors are found in the CDRs, these regions are most variable. Furthermore, while CDR1 and CDR2 are found in the variable (V) region, CDR3 includes some of V, all of diversity (D of heavy chains only) and joining (J) regions. So, CDR3 is the most variable and can represent the most timely immune reaction events and panorama of the immune system.

Accordingly, we have revised the description of CDR3 for why we used these sequences to represent the timely immune reaction to the renal transplantation and allied events (Page 2 Line 18-23). For more clear description, we cited two more references here (Ref 20, 21).

On the other hand, regarding the normalization of these data, we have described the detailed calculation procedure in the methodology and explained the reason for using these methods to normalize or compare data from heterogenous patients in our original as well as revised submissions. These statements are in Page 6 (Section 2.4, 1st paragraph) and are also shown below:  

“A very convenient feature of the iRweb software is CDR3 algebra, which allows the comparison of the CDR3 sequences from one data set to another data sets for identifying shared CDR3 profiles. This allows for a comparison among samples of different time points during the monitoring of disease fluctuation. All CDR3 frequencies were artificially scaled to 10 million reads to account for differences in read depth among samples, making comparisons between samples easier with this normalization step.”

Conflicting statements. ‘There was no single BCR IGH CDR3 sequence being exhibited in all 14 samples. However, there was a shared profile?  Can you please complain what the shared profile means? Define how to get something shared, when none of the sequences are the same.

Ans:  We are very sorry not to clearly explain these experimental results. Actually, every single patient exhibited his or her unique sequence in V, D, and J regions as a whole but there was a short profile of nucleotides that are the same in sequence being expressed in some of the samples (7 out of 14) at the baseline. So, we changed this paragraph to “There was no single whole BCR IGH CDR3 sequence expressed in all 14 patients. However, there was a profile of nucleotides containing 544 IGH CDR3 sequences expressed across some of the patients as listed in Table S2. “ARDLDY” is this shared CDR3 sequence being expressed in 7 out of 14 baseline samples” (Page 7, Line 36-39)

There are still English grammar mistakes throughout the manuscript.

Ans: We are very sorry to have these mistakes which led to the unclear explanation of our experimental results because English is not our mother tongue. We would like to subject our manuscript to the grammar correction provided by the Editorial Office of JCM. We, the authors, are obligated for the payment for this service.

The clinical phenotype in patients 13 has yet to be examined.  Why not? This would help increase the ability to understand what is happening in this patient since the sample size here is very small. Are the data unavailable? Should this patient be excluded from the study for clarity of the research?

Ans: We are very sorry for the incompleteness of the data provided. We have revised our supplement Table 1 with addition of more data to make it more complete and the sample numbers are replaced with patient numbers to avoid confusion. The revised data are highlighted red in a separate file for your reference.

Unfortunately, the patient 13 you mentioned was actually patient 9 (designated as sample 13) who died during the study course and her data were thus incomplete. Because this patient committed suicide sooner after she developed TCMR, further data could not be obtained. We have described this situation in the original submission as well revised submission (Page 11 Figure 1 legends and Page 18 Line 47 in supplement legends). Since her data in the former part of the study are still complete and deserve analysis, we did not give up them and have retained them in revised manuscript since our cohort is small. The data we added in table S1 (mainly pt 13 and pt 14), were designated as sample No. 18 and No. 19 originally. We have replaced them with the patient number.

Although the authors provide some explanation of why serum creatinine may not correlate with iR, more explanation is needed.  Since creatinine is a direct measure of kidney function, the explanation suggests that there is no relevance of iR in patients. 

Ans: Thank you for these important suggestions. We completely agree with you that creatinine is the direct measure of kidney function rather than DI, D50 or entropy here in the manuscript. Our actual purpose is to describe that diversity changes may happen before the adverse event and subsequent creatinine elevation rather than saying that creatinine is insensitive. We are very sorry to wrongly state these. So, we have revised the text as “The serum creatinine levels of these patients at different times were correlated to the adverse events as well as fluctuations of D50/DI/H. Although creatinine levels were correlated well to the adverse events, the times of D50/DI/H changes seemed coming before creatinine change although they were not statistically verified. This implies that personal iR indices may be also regarded as good indicators for post-transplant monitoring of immune status.” (Page 9, Line 34-38) and in Discussion as “Indeed, the serum creatinine levels or clinical renal function did show significant relationship with the adverse events or graft failure, iR change seemed occurring before the events and creatinine changes which were usually happening after renal functions were substantially disturbed (data not shown). This is probably because the iR change is the driver of body’s defense against extrinsic offending events and serum creatinine change is the outcome of body’s defense mechanism. So, the level of creatinine is less likely to exhibit conspicuous fluctuations in such a short time of monitoring (e.g., one or two weeks in a case of chronic rejection). Alternatively, the iR changes may play as a whistler to harbinger the favorable or adverse events if more data are accumulated.  (Page 16, Line 9-18)

Reviewer 2 Report

Manuscript title

High Throughput Sequencing of Complementarity Determining Region 3 in Heavy Chain of B-cell Receptor in Renal Trans- plant Recipients: A Preliminary Report

General comments:

The authors have assembled a small and diverse cohort of renal transplant patients who exhibit a different baseline risk and clinical course post-transplant. Longitudinal BcR repertoire sequencing was performed. The strengths of this manuscript include the fact that BcR sequencing is still at a very early stage of investigation in renal transplantation, and this is an interesting descriptive manuscript with tracking of diverse clinical outcomes although the cohort is quite small. The methodology for BcR sequencing is sounds, although no innovations on methodology were made. The weaknesses of this manuscript lie in the heterogeneity of courses and outcomes (many of which are not well defined), the small cohort of subjects, the enormous heterogeneity in B-cell clonal patterns observed and the different BcR repertoire metrics with clinical outcomes. Moreover, important immune aspects including HLA epitope/eplet mismatch and development of de novo DSA were omitted. Last, several areas of BcR repertoire sequencing analysis were not explored including V gene usage and class-switching. The article has many positive features, and if revised and re-structured would be of interest to the field.

Abstract

  1. The abstract needs to be reviewed for grammar and syntax.
  2. Some statements are tautologous – “..graft failure has an immense effect on allograft survival ..”
  3. The abstract does not indicate how many patients were included in this study
  4. The conclusion statement does not follow from the results section necessarily.
  5. It is impossible to understand from the complex data presented what happened in which patients and how (if) it correlated with clinical outcomes over time.

Introduction

  1. Generally the introduction needs to be revisited for grammar and syntax.
  2. The introduction lacks clarity as the authors fail to distinguish between cell-mediated and antibody-mediated rejection. Specifying which immune outcome of they are discussing would strengthen the introduction.
  3. There are at several studies characterizing the B cell immune repertoire in transplant. The introduction would benefit from a more clear summary of this and how the present study fits into this context
  4. The authors describe “subtle changes within the immune system that predispose rejection episodes”. What doe they mean by this outcome of interest?
  5. Many of the statements are not yet substantiated, or supported only by small data reports. For example, the statement “Patients who develop rejection demonstrate a specific set of clonal expansion that will persist after the rejection”. This is far from accepted or proven.

Methods

  1. In section 2.1 the authors describe their patient population (ie. Patient #1 and #6 were transplant patients with blood type mismatch etc.). It is extremely difficult to interpret the patients forming each sub-group in the results and this must be corrected before publication. This data would be best moved to the results section and precise groups defined in Table 1. Moreover, they used the term “presented with class I and class II PRA” to describe two of the patients. Given that they received IVIg and rituximab presumably the authors mean that they have crossed antibodies in these patients, but this would benefit from clarification.
  2. The authors have selected 4 high risk patients for this study (ie. crossing ABO antigens and crossing donor specific antibodies). Given that it is not routine practice to do this, the manuscript would be strengthened by including an explanation for why these patients were selected.
  3. The authors did not explain what diagnostic criteria were used for the various infectious and rejection outcomes in the patient cohort.
  4. Good description of the sequencing methodology
  5. The authors describe 3 methods of evaluating repertoire diversity including D50 (measures diversity of the top 50% of CDR3 sequences by frequency), DI (sum of the frequency of a given clone divided by the sum of frequencies squared), Shannon entropy. PCA analysis and hierarchical clustering was used to evaluate degree of similarity between groups. Why have the authors chosen 3 different diversity measure?

Results

  1. Generally, the results would benefit from revision for grammar and syntax.
  2. Table 1 describes basic demographic features of the cohort and Table 2 their clinical course. It would be helpful to include what kind of induction the patients had.
  3. Table 2 is arranged in a confusing way, and would benefit from some reorganization for clarity.
  4. Findings include generally a rise in BcR diversity after transplant except in patients who received rituximab. Good correlation between different measures of diversity (internal consistency is good).
  5. Paragraph 2 of 3.1 describes major clinical findings alongside measures of diversity. While it is a fundamentally important paragraph for this paper, it lacks clarity as there is little structure. It is therefore difficult to understand what clinical events result in changes in BcR repertoire diversity.
  6. The authors last sentence in paragraph 3.1 states ‘serum creatinine or clinical renal function is also not a good indicator for post-transplant monitoring of immune status.’ This is based on the feact that BcR CDR3 sequence diversity does not correlate with serum creatinine and appear not to specifically track with the described clinical events. First, this perhaps belongs in the discussion. Second, this is a very contentious statement given that creatinine and biopsy are the current gold standard of transplant monitoring, and the limited data the authors present to show that BcR sequencing is in fact an improvement over this.
  7. Figure 1 would benefit from including all treatments and not just rituximab. Eg. patients 7, 9, 14 all had TCMR and a reduction in D50 at time point 1. Is this reduction in D50 because they received steroids as part of their rejection treatment?
  8. Figure 2A uses sample numbers instead of patient numbers, this is very confusing since the sample numbers are different than the patient numbers but in the same range. Please convert to patient numbers.
  9. Based on PCA and hierarchical clustering of CDR3 sequences, the authors divide the population into 3 groups at baseline and attribute this to their ESRD. Could this be due to other factors like infectious exposure? Other conditions? How does this related to risk of rejection or transplant complication?
  10. In section 3.3 the authors describe movement of each patient in the PCA plot, but it is unclear what this means. It is comforting that patients PCA profiles tend to cluster together. Samples 11 (patient 7), 17 (patient 12), and 3 (patient 3) are markedly different along the PCA axis. What is different about these patients? They all appear to have different clinical courses. 2 have infections and rejection, whereas one has no post-transplant clinical events.
  11. In figure 3B and 3C the authors indicate there are 3 clusters. This is unclear from the figure.
  12. V gene usage and class switching is another common outcome measured in BcR sequencing. Why did the authors chose to omit these from their data analysis?

Discussion

  1. The discussion would benefit from revision for grammar and syntax.
  2. The authors state that the BCR IGH datasets and CDR2 count listing “reflected the changes in immune status along the whole renal transplant course”. It is impossible to know what this means. A close reading of the manuscript shows that they divide the patients into 3 clinical groups (Ritux induction, TCMR, the rest), these however do not correlate with the hierarchical clusters from figures 2 and 3, nor do the patterns of diversity appear the same when charted over time and corelated with immune events. Please clarify how the BcR sequencing results track with clinical and immune outcomes.
  3. BcR sequencing would presumably be of notable interest in AMR. The one patient with AMR was not discussed at length in the paper, but showed increased diversity and not decreased diversity with AMR - please explain. DSA data was not provided for any of these patients and may be an interesting adjunct clinical data point to correlate with BcR repertoire. What about HLA eplet/epitope mismatch? How does this play into
  4. Shannon entropy does not change over time as much as the other two measures and therefore may not be a good marker of immune change in renal transplant patients. Based on how this measure is calculated why do you think this is the case?

Author Response

Reviewer 2

The authors have assembled a small and diverse cohort of renal transplant patients who exhibit a different baseline risk and clinical course post-transplant. Longitudinal BcR repertoire sequencing was performed. The strengths of this manuscript include the fact that BcR sequencing is still at a very early stage of investigation in renal transplantation, and this is an interesting descriptive manuscript with tracking of diverse clinical outcomes although the cohort is quite small.

Ans: Thank you for your positive comments. Actually, it was very difficult to collect this cohort of renal transplant patients because in our country, the donors are very scarce and most of the patients are reluctant to participate in studies such like this because of personal as well as family factors. So, we have added a subtitle “A preliminary report” to the Title of this article both in original and revised submissions. Indeed, although the cohort is small, we think these data are precious and could provide some incentive for further investigations to panoramically understand the immune responses in renal transplantation.

The methodology for BcR sequencing is sounds, although no innovations on methodology were made. The weaknesses of this manuscript lie in the heterogeneity of courses and outcomes (many of which are not well defined), the small cohort of subjects, the enormous heterogeneity in B-cell clonal patterns observed and the different BcR repertoire metrics with clinical outcomes.

Ans: Thank you for your comments. We agree that the clinical course and outcome of the patients are heterogeneous since the cohort is small as has also been mentioned by the reviewer 1 above. Because of this, we carefully stratified our patients into rituximab, DCD, TCMR, and tolerance groups for more precise comparison of so many indices. This inevitably reduced the number in each group and made statistics more difficult to perform. We have stated these paucities and limitations in the Conclusion (Page 18 Line 29-39) as “There were quite inevitable defects in the present investigation, i.e., difficult to recruit the patients because of the low incidence of renal transplantations and low consent rate of enrollments in this country as well as higher costs of individual iR sequencing. The small number of enrollments directly resulted in the heterogeneity of stratified groups. Moreover, V gene usage and class-switching of the immunoglobulins were not explored. Although the experimental procedures were quite preliminary and complex, we propose that further investigations in this field may help establish a convenient and reliable immunogenetic biomarker to be used as a good tool for following up the clinical course of individual renal transplant patients. The way to interpret various responses in individual renal transplant patient using the iR information deserves further investigation with a larger sample size.”

Moreover, important immune aspects including HLA epitope/eplet mismatch and development of de novo DSA were omitted.

Ans: In our cohort, we only have one patient (No. 13) with very low titer of (equivocal) de novo DSA development (shown in revised Table S1). However, the patient clinical course was very smooth except that at the initial post-transplant stage, there were two episodes of Klebsiellae UTI (Table 2). The clinical status and renal function are currently stable. In the figure 1b, it also shows stable of the diversity in the late stage of follow-up period.

HLA mismatch was assessed in every patient and has been listed in Table 1 for all of them.

Last, several areas of BcR repertoire sequencing analysis were not explored including V gene usage and class-switching.

Ans: We are very sorry for this. Because of budget limitation, we were unable to explore this field since NGS is quite a “budget burning” experiment. We have also mentioned these weak points in our Conclusion (Page 18, Line 33).

The article has many positive features, and if revised and re-structured would be of interest to the field.

Ans: Thank you for your appreciation and encouragement. In addition to adding more data that reviewers suggested, we also extensively revised our manuscript grammatically and we have highlighted these changes in red in the traction change version of the revised manuscript. However, we are still eager to have our revised manuscript be rechecked for grammar by the Editorial Office of JCM.

On the other hand, because of the small cohort as well as immediacy of the sampling, we cannot collect samples retrospectively at this time point to recapitulate the events. We are very sorry about this. We have reiterated the importance of future studies to further understand the immune repertoire change in response to transplantation (Page 18 Line 29-39).

Abstract

  1. The abstract needs to be reviewed for grammar and syntax.

Ans: Thank you for the comments. We would like to let the entire manuscript be revised for grammar and syntax by the Editorial Office. We’ll pay for your kind help.

  1. Some statements are tautologous – “…graft failure has an immense effect on allograft survival ..”

Ans: Thank you for the comments. To avoid tautologuous statements, we have deleted the redunlant sentence “graft failure has an immense effect on allograft survival…” and changed it to “Graft failure resulted from rejection or any other adverse event is usually originated from aberrant and/or exaggerated immune response and is often catastrophic in renal transplantation.” (Page 1, Line 21-23)

  1. The abstract does not indicate how many patients were included in this study

Ans: Very sorry for the inadvertence. We have added the statement as “We monitored B cell receptor (BCR) complementary determining region 3 (CDR3) immunoglobulin heavy chain (IGH) sequence in immune repertoire (iR) of 14 renal transplant patients using next generation sequencing (NGS), correlating its diversity to various clinical events occurring after transplantation.” in abstract (Page 1 Line 24-27)

  1. The conclusion statement does not follow from the results section necessarily.

Ans: We have extensively rewritten the conclusion as “Clonal diversity in BCR IGH CDR3 varied depending on clinical courses of 14 renal transplant patients including B cell suppression therapy, TCMR, DCD, and graft tolerance. Adverse events implicated in renal graft might lead to different clustering of BCR iR. However, these preliminary data need further verification in further studies for the possible applications of iR changes as genetic expression biomarkers or laboratory parameters to detect renal graft failure/rejection earlier.” (Page 1 Line 39-44) to make the Results and Abstract Conclusions consistent.

  1. It is impossible to understand from the complex data presented what happened in which patients and how (if) it correlated with clinical outcomes over time.

Ans: We are sorry for this since the patients’ courses were varied and diverse, we could not accommodate so much data in a limited space. However, we have tried our best to rewrite the conclusion as “Clonal diversity in BCR IGH CDR3 varied depending on clinical courses of 14 renal transplant patients including B cell suppression therapy, TCMR, DCD, and graft tolerance. Adverse events implicated in renal graft might lead to different clustering of BCR iR. However, these preliminary data need further verification in further studies for the possible applications of iR changes as genetic expression biomarkers or laboratory parameters to detect renal graft failure/rejection earlier.” (Page 1, Line 39-44). Also, we have presented chronological D50/DI/H changes in figures (Fig. 1) as correlated to the individual clinical events (Table 2). This study is not intending to establish a predicting tool for the outcome of renal transplantation. Rather it is just a preliminary report showing that iR diversity change is parallel to the host response to renal graft and other confounding factors. After more extensive studies, this tool may be used in personal medicine for individual renal transplant patients.

 Introduction

  1. Generally the introduction needs to be revisited for grammar and syntax.

Ans: We would like our manuscript be revised by your editorial office. We are obligated with the payment.

  1. The introduction lacks clarity as the authors fail to distinguish between cell-mediated and antibody-mediated rejection. Specifying which immune outcome of they are discussing would strengthen the introduction.

Ans: Thank you for the suggestion. The AMR involves B cell immune functions. We have rearranged our references (Ref 17, 19) regarding the T cell iR and B cell iR in renal transplantation in this paragaraph and rewrote several sentences as (Page 2, Line 11-30) “Presumably, sequencing the expressed T or B cell….rejection and there did have such reports [17-19]. A characterization …. revealed that patients … rejection [18]. Also, patients who…. Most sequence variation associated with immunoglobulin’s (Ig’s) and T cell receptors (TCRs) are found in the complementary determining regions (CDRs), which are most variable [20]. CDR1 and CDR2 are found in the variable (V) region, but CDR3 includes some of V, all of diversity (D) region of heavy chains and joining (J) regions [21]. So, CDR3 is the most variable and can more reliably indicate the most timely immune reaction events and immune status in transplant patients. In the present … in post-transplant time. Although … has been reported previously [19], .. tried to apply mathematical calculation mode to analyze the immune diversity change quantitatively. We …. episodes.

  1. There are at several studies characterizing the B cell immune repertoire in transplant. The introduction would benefit from a more clear summary of this and how the present study fits into this context

Ans: Thank you for the suggestions. We have rearranged our previous references (mainly ref 18) for the description and made an inference in Introduction (Page 2, Line 7-17) and Discussion (Page 15, Line 18-21) (Ref. 17-19, 33, 34).

  1. The authors describe “subtle changes within the immune system that predispose rejection episodes”. What doe they mean by this outcome of interest?

Ans: Sorry for the ambiguity of the sentence. We have revised it to “We are interested to see if B cell iR could provide alarms in advance for adverse clinical events including rejection within the immune system.” (Page 2, Line 29-30)

  1. Many of the statements are not yet substantiated, or supported only by small data reports. For example, the statement “Patients who develop rejection demonstrate a specific set of clonal expansion that will persist after the rejection”. This is far from accepted or proven.

Ans: Sorry for the ambiguity of description. Actually, this has been mentioned in previous publication and we have cited this paper [18]. For convenience and concise, we have condensed this paragraph to a sentence as “…patients who develop rejection have a more diverse iR before transplantation, suggesting a predisposing liability to rejection, and also demonstrate a specific set of clonal expansion that will persist after the rejection [18]. (Page 2 Line 13-17)

 Methods

 In section 2.1 the authors describe their patient population (ie. Patient #1 and #6 were transplant patients with blood type mismatch etc.). It is extremely difficult to interpret the patients forming each sub-group in the results and this must be corrected before publication. This data would be best moved to the results section and precise groups defined in Table 1. Moreover, they used the term “presented with class I and class II PRA” to describe two of the patients. Given that they received IVIg and rituximab presumably the authors mean that they have crossed antibodies in these patients, but this would benefit from clarification.

Ans: Thanks for the suggestions. For the sake of clear description of the enrollments, we should put the detailed patients’ clinical data in the Methodology. In order to make the description of subgroups of patients more clearly in the Results section, we reiterated these stratifications of patients pertinently and more concisely in this Section(3.1.) as “Patients #1 and #6 had blood type mismatch. Patient #1 and #4 had PRA-I or II but not DSA. So, we gave preventive desensitization therapy with rituximab and IVIG before transplantation to deplete B-cells in these 3 patients. Patient #8 presented with low PRA-I titer but was O in blood type, compatible with the donor, and only had 2 HLA mismatches. So, he didn’t undergo rituximab.” (Page 7, Line 16-20). For more clear description of precise groups, we added in Table 1 a right column “remark” designating subgroup as DS (desensitization), DCD (donation after cardiac death), TCMR (T cell mediated rejection) and blank (tolerance or no specific events).

  1. The authors have selected 4 high risk patients for this study (ie. crossing ABO antigens and crossing donor specific antibodies). Given that it is not routine practice to do this, the manuscript would be strengthened by including an explanation for why these patients were selected.

Ans: Thanks for your comments. Actually, we have described the reason for including these patients in the Methodology and Results (Section 2.1. and 3.1.). Because of the main aim of the present study is to see if B cell iR is changed after transplantation, inclusion of these cases would be more significant for our purpose.

  1. The authors did not explain what diagnostic criteria were used for the various infectious and rejection outcomes in the patient cohort.

Ans: The definition of rejection or any other events were according to the pathology, infections were according the bacterial cultures and pathology and viral infection or BKV were confirmed by cytology or pathology in the case of BK viral infection or PCR in the case of CMV. These descriptions are added in the footnote of Table 2 as “The individual bacterial infections were verified by culture results, CMV infection was confirmed by polymerase chain reaction, TCMR, AMR or PVN was confirmed by histopathology and BKV was confirmed by decoy cells in urine cytology.”

  1. Good description of the sequencing methodology

Ans: Thank you for your comment.

  1. The authors describe 3 methods of evaluating repertoire diversity including D50 (measures diversity of the top 50% of CDR3 sequences by frequency), DI (sum of the frequency of a given clone divided by the sum of frequencies squared), Shannon entropy. PCA analysis and hierarchical clustering was used to evaluate degree of similarity between groups. Why have the authors chosen 3 different diversity measure?

Ans: the purpose of these choose was to compare 3 different methods for selection of the best and sensitive method for follow-up of the patients’ clinical condition as have been shown in Figure 1 and we have retained these descriptions in the Discussion.

Results

  1. Generally, the results would benefit from revision for grammar and syntax.

Ans: Thank you for your comment. The manuscript will be subjected for your revision of grammar and syntax as mentioned above.

  1. Table 1 describes basic demographic features of the cohort and Table 2 their clinical course. It would be helpful to include what kind of induction the patients had.

Ans: This question is the same as the above. We have added respective details in the appropriate sites.

  1. Table 2 is arranged in a confusing way, and would benefit from some reorganization for clarity.

Ans: Thank you for the comment. Actually, the event occurrence time has been designated in the Table 2 and explained in the footnote as “Column T1 means period between T0 and T1, T2 means period between T1 and T2, and so on.”

  1. Findings include generally a rise in BcR diversity after transplant except in patients who received rituximab. Good correlation between different measures of diversity (internal consistency is good).

Ans: Thank you for the comment.

  1. Paragraph 2 of 3.1 describes major clinical findings alongside measures of diversity. While it is a fundamentally important paragraph for this paper, it lacks clarity as there is little structure. It is therefore difficult to understand what clinical events result in changes in BcR repertoire diversity.

Ans: Thank you for the comments. We have revised our description in 2.1. and 3.1. as have been mentioned above. We have reiterated the courses of these patients in Results (3.1.)

  1. The authors last sentence in paragraph 3.1 states ‘serum creatinine or clinical renal function is also not a good indicator for post-transplant monitoring of immune status.’ This is based on the feact that BcR CDR3 sequence diversity does not correlate with serum creatinine and appear not to specifically track with the described clinical events. First, this perhaps belongs in the discussion. Second, this is a very contentious statement given that creatinine and biopsy are the current gold standard of transplant monitoring, and the limited data the authors present to show that BcR sequencing is in fact an improvement over this.

Ans: Thanks for your comments and we apologized for this contentious statement and we have revised our descriptions to a more conservative tone as have been mentioned above.

  1. Figure 1 would benefit from including all treatments and not just rituximab. Eg. patients 7, 9, 14 all had TCMR and a reduction in D50 at time point 1. Is this reduction in D50 because they received steroids as part of their rejection treatment?

Ans: Since the purpose of Figure 1 is not to show the correlation of treatments to the fluctuation of diversity, we did not show treatment scheme in the figures. The reason that we added rituximab in Figure 1A is because of the subgrouping these patients as desensitization with B cell suppression, directly relevant to BCR repertoire. For the treatment of TCMR, we did not add extraordinary dose of steroid to ameliorate the condition. Moreover, the changes in diversity came before the events whereas the treatments were modified after the events.

  1. Figure 2A uses sample numbers instead of patient numbers, this is very confusing since the sample numbers are different than the patient numbers but in the same range. Please convert to patient numbers.

Ans: We are very sorry for this. However, because the figures are actually the original data taken from the machine, we cannot change them at this time. We added designation of each patient to individual sample number in the figure legends.

  1. Based on PCA and hierarchical clustering of CDR3 sequences, the authors divide the population into 3 groups at baseline and attribute this to their ESRD. Could this be due to other factors like infectious exposure? Other conditions? How does this relate to risk of rejection or transplant complication?

Ans: We agree with the reviewer’s point that these may be due to rejection or infection or any other complication. These were exactly the limitation of this preliminary report, we are currently still unable to interpret the specific fluctuation to a specific event, which need further future in-depth investigations with larger number of samples. We have mentioned this defect of the investigation in our Conclusion (Page 18, Line 26-28).

  1. In section 3.3 the authors describe movement of each patient in the PCA plot, but it is unclear what this means. It is comforting that patients PCA profiles tend to cluster together. Samples 11 (patient 7), 17 (patient 12), and 3 (patient 3) are markedly different along the PCA axis. What is different about these patients? They all appear to have different clinical courses. 2 have infections and rejection, whereas one has no post-transplant clinical events.

Ans: Thank you for your comments. These are exactly the defect of our present investigation. With limited data we could not exactly identify the true reasons for these changes. We have given some possible explanation for this (PC1 and PC2 data ….. Patient #3 exhibited a slight consistent horizontal shift along the PC1 axis whereas Patient #7, #12 and #13 showed a slight consistent vertical shift along the PC2 axis. Interestingly, two of them (patients #7, #12) exhibited a final fall-down of D50/DI at the end of the study. Both of them suffered from TCMR. The correlation between …….in patient #13 has yet to be examined with more clinical biomarkers. AHC analysis confirmed there were several (at least 2) clusters formed (Fig. 3b & c), indicating that there might have several (at least 2) shared mechanisms to help maintain graft tolerance. (Page 13, Line 16-25). However, these are just a hypothesis. With large number of enrollments, we hope that our future studies will give an answer to this critical question.

  1. In figure 3B and 3C the authors indicate there are 3 clusters. This is unclear from the figure.

Ans: We are very sorry to describe these results contentiously. We now revise them with more conservative manner as “ … there were several (at least 2) clusters formed. …. Have several (at least 2) shared mechanisms to …..” (Page 13, Line 23-25)

  1. V gene usage and class switching is another common outcome measured in BcR sequencing. Why did the authors chose to omit these from their data analysis?

Ans: We are very sorry to omit these analyses because of budget limitation (the tests are very expensive). We have mentioned this in the conclusions (limitation of the study) (Page 18 Line 33).

Discussion

  1. The discussion would benefit from revision for grammar and syntax.

Ans: thank you for your comment. The manuscript will be subjected for your revision of grammar and syntax as mentioned above

  1. The authors state that the BCR IGH datasets and CDR2 count listing “reflected the changes in immune status along the whole renal transplant course”. It is impossible to know what this means. A close reading of the manuscript shows that they divide the patients into 3 clinical groups (Ritux induction, TCMR, the rest), these however do not correlate with the hierarchical clusters from figures 2 and 3, nor do the patterns of diversity appear the same when charted over time and corelated with immune events. Please clarify how the BcR sequencing results track with clinical and immune outcomes.

Ans: Regarding the desensitization group (Rituximab & IVIG), CDR3 iR diversity remained low (showing in Fig 1a), regarding the TCMR group, the diversity went down before rejection and persisted for a while until recovery (showing in Fig 2c), and regarding the tolerance group, the diversity jumped up very quickly after grafting (showing in Fig. 1b). All of these indicated the the BCR IGH CDR3 iR reflected the changes in immune status along the transplant course. We have discussed this in detailed in the Results and Discussion. On the other hand, the hierarchical clustering was aimed to monitor the 1st and 2nd most frequently expressed clones of genes in respective patients at respective time. They were not aimed to monitor the overall diversity of iR in individual patients. Furthermore, the Figure 1 is aimed to show the dynamic changes along the time course after transplantation. It is not aimed to show the correlation between clinical events and diversity changes. We have put this point of view in Discussion (limitation of the study and conclusion).  

  1. BcR sequencing would presumably be of notable interest in AMR. The one patient with AMR was not discussed at length in the paper, but showed increased diversity and not decreased diversity with AMR - please explain. DSA data was not provided for any of these patients and may be an interesting adjunct clinical data point to correlate with BcR repertoire. What about HLA eplet/epitope mismatch? How does this play into

Ans: Patient 14 developed TCMR and then a suspected microvascular inflammation in the graft. As this lesion was only equivocal, the diagnosis of AMR was only pending. In addition, there was no de novo PRA throughout the whole follow-up time (Clinically, this patient did not have any symptoms suggestive of the development of DSA). We concluded that it was not AMR. A new Table 2 is replaced of old Table 2 to more clearly describe this. These were recorded additionally in revised Table S1 in addition to revised Table 2. On the other hand, only one patient (# 13) developed a very mild de novo PRA at the T3, which could be reflected by a very tiny dip in the Diversity as shown in Fig 1b right panel. There was no clinical symptom suggestive of de novo DSA development. These were described in the Results and Discussion (Page 9, Line 26-29, Page 16, Line 37-39 and Page 17, Line 13-16)

  1. Shannon entropy does not change over time as much as the other two measures and therefore may not be a good marker of immune change in renal transplant patients. Based on how this measure is calculated why do you think this is the case?

Ans: As indicated in the Figure 1, the Shannon entropy curve lies flat in almost all of the case throughout the courses which are the strong evidence of its irrelevance to the clinical course. We have selected the most remarkable changes of these values to compare and found there was not statistically significant difference between these values [entropy (11.31+1.60 at T0 and 11.04+3.50 at T1, P = 0.706) (Page 7, Line 30-31).